# Snapshot Quantitative Phase Imaging with Acousto-Optic Chromatic Aberration Control

**DOI:** 10.3390/s25144503

**Published:** 2025-07-20

**Authors:** Christos Alexandropoulos, Laura Rodríguez-Suñé, Martí Duocastella

**Affiliations:** 1Department of Applied Physics, Universitat de Barcelona, C/Martí i Franquès 1, 08028 Barcelona, Spain; chrisalexandropoulos@ub.edu (C.A.); laura.rodriguez.sune@ub.edu (L.R.-S.); 2Institut de Nanociència I Nanotecnologia (IN2UB), Universitat de Barcelona, Av. Diagonal 645, 08028 Barcelona, Spain

**Keywords:** label-free microscopy, acousto-optics, transport of intensity equation, liquid lens, illumination encoding

## Abstract

The transport of intensity equation enables quantitative phase imaging from only two axially displaced intensity images, facilitating the characterization of low-contrast samples like cells and microorganisms. However, the rapid selection of the correct defocused planes, crucial for real-time phase imaging of dynamic events, remains challenging. Additionally, the different images are normally acquired sequentially, further limiting phase-reconstruction speed. Here, we report on a system that addresses these issues and enables user-tuned defocusing with snapshot phase retrieval. Our approach is based on combining multi-color pulsed illumination with acousto-optic defocusing for microsecond-scale chromatic aberration control. By illuminating each plane with a different color and using a color camera, the information to reconstruct a phase map can be gathered in a single acquisition. We detail the fundamentals of our method, characterize its performance, and demonstrate live phase imaging of a freely moving microorganism at speeds of 150 phase reconstructions per second, limited only by the camera’s frame rate.

## 1. Introduction

Multiple microscopy methods now exist for rapidly extracting quantitative phase information from samples exhibiting poor optical contrast, eliminating the need for staining or fluorescent labeling. Among them, the transport of intensity equation (TIE) can be advantageous be advantageous due to its ability to reconstruct phase values from incoherent light intensity without requiring complex interferometric setups [1]. Typically, only two or more axially displaced images suffice to reconstruct a sample’s phase map [2]. While iterative phase retrieval algorithms such as the Gerchberg–Saxton (GS) method can be applied in similar configurations, they are computationally intensive, sensitive to noise, and less suited for real-time imaging [3]. In contrast, TIE enables non-iterative phase reconstruction from axial intensity gradients, making it more computationally efficient, robust to noise, and overall, more appropriate for dynamic biological imaging. However, most TIE implementations require mechanical translation of the sample or objective to obtain the defocused images. This limits the acquisition speed to some dozens of hertz, a speed insufficient for characterizing dynamic processes. Efforts to address this issue include systems capable of capturing the defocused images in a single snapshot, such as those based on using different optical elements to simultaneously project the displaced images into one or multiple cameras [4,5,6,7,8,9,10], or exploiting chromatic aberrations with white illumination and a color camera [11]. Unfortunately, these methods are normally designed to operate with a fixed axial displacement between images. Note that adjusting these inter-plane distances is necessary to optimize phase reconstruction accuracy [2].

Here, we report a TIE system that combines user-adjustable defocusing with snapshot phase retrieval. Our method is based on coupling a fast varifocal lens and a multi-colored pulsed light source into a commercial microscope for rapid chromatic aberration control. Specifically, we used a Tunable Acoustic Gradient (TAG) index of refraction lens (TAG lens, TAG Optics Inc.) for continuous axial focus scanning at microsecond timescales [12,13]. By using light pulses much shorter than the TAG lens scanning time, it is possible to capture information from a single axial plane within the scanned range. In this case, adjusting the time delay between pulses and axial scanning allows fast control of the focus plane. In a previous implementation, we exploited this unique feature to sequentially capture three axially displaced images for TIE-based phase reconstruction [14]. However, the requirement of three acquisitions per phase map effectively reduced quantitative phase imaging (QPI) rates to one-third of the camera’s frame rate. Building on these results, we present herein a new architecture in which three pulsed sources of distinct colors are used for illumination, each synchronized with a different time delay relative to the TAG lens scanning. This enables precise adjustment of the focal position for each color, with the TAG lens acting as a chromatic aberration control unit. Following the acquisition with a color camera and color-channel splitting, the three images needed for TIE can thus be gathered in a single snapshot. While similar TAG-enabled systems incorporating multi-color illumination and snapshot information retrieval have previously been developed for applications such as multi-plane bright-field microscopy [15] and extended axial-range single particle tracking [16], this represents the first implementation of such a setup in the context of quantitative phase imaging (QPI). This novel application leverages the advantages of rapid axial scanning and simultaneous multi-plane acquisition inherent to TAG-based approaches, enabling high-speed, label-free imaging of transparent biological samples with quantitative contrast—capabilities that were not previously demonstrated using this technology.

## 2. Materials and Methods

The chromatic aberration-controlled TIE microscope was implemented in a commercial inverted system (Nikon ECLIPSE Ti2). It featured a commercial 4-pin RGB-LED (L5RGB/4, Diotronic, λ_Red_ = 625 nm, λ_Green_ = 525 nm, λ_Blue_ = 470 nm), a 20 x, 0.50 NA objective (Nikon Plan Fluor), a TAG lens, and a color camera (DFK 37AUX287, The Imaging Source), as shown in Figure 1a. The RGB-LED replaced the standard white lamp used for transmitted illumination, while the TAG lens and color camera were mounted on the left port of the commercial microscope. Two lenses with a focal length of 200 mm each were placed between the TAG lens in a 4-f configuration. This setup ensured that the TAG lens was positioned at a plane conjugate to the back pupil of the objective lens, thus avoiding magnification changes across different axial positions. A pulse delay generator (DG645, Stanford Research Systems) controlled the driving signals and time delays for each individual color. In more detail, we selected a pulse duration of 500 ns for each color, triggered at the same rate as the TAG lens driving frequency (142 kHz)—the pulse duration was about one order of magnitude shorter than the lens oscillation period. No synchronization was used between light pulses and the color camera. Thus, each acquired image consisted of the integration of multiple pulses, the number of which depended on the desired frame rate, and therefore, exposure time. By using multiple pulses for each image, we achieved a significant enhancement in signal-to-noise ratio (SNR).

Importantly, the broad spectral response of the different channels in a color camera, combined with the wide emissions of commercially available LEDs, can cause each camera color channel to receive signals not only from the intended LED color but also from adjacent colors. This phenomenon, known as color leakage, can lead to significant errors in phase estimation. To compensate for this effect, we employed a well-known correcting scheme that assumes the magnitude of color leakage in each channel to increase linearly with the light intensity of the other colors [17,18,19]. Specifically, we modelled each color channel captured by the camera as the product of the camera’s response to different colors—represented by the camera response matrix—and the true color channels of the image, which can be written as:(1)IRcamIGcamIBcam=RRRRGRRBRRRGRGGRBGRRBRGBRBBIRimgIGimgIBimg 
where $I_{i}^{\mathrm{cam}}\;and\;I_{i}^{\mathrm{img}}$ represent the camera intensity and image intensity at the i-th color channel (i = red, green or blue), respectively, and $R_{j}^{i}$ denotes the camera response at channel *i* when illuminated with the LED of color j (j = red, green or blue). Therefore, it suffices to acquire three images of a sample-free area using one individual color at a time to calculate the camera response matrix. By calculating its inverse and multiplying it with the images captured with the camera, the true color channels of a sample image can be obtained, which yields to:(2)IRimgIGimgIBimg=RRRRGRRBRRRGRGGRBGRRBRGBRBB−1IRcamIGcamIBcam

As shown in Table 1, this process reduced color leakage to less than 3%, making the captured images suitable for QPI.

A fast Fourier transform (FFT)-based solver was used to compute the transport of intensity equation (TIE) due to its speed and efficiency. The method requires in-focus and defocused images, defocus distance, pixel size, and wavelength as inputs. A minimum intensity threshold is applied to avoid numerical errors. The optimal defocus distance for each sample was selected based on spatial frequency content and detector noise, following established criteria [14]. The implementation of the TIE solver and all image processing was performed in a regular desktop computer featuring an Intel Core i7 processor (10700 CPU @ 2.90 Ghz) and 16 GB RAM. All algorithms were implemented using MATLAB R2019B.

## 3. Results and Discussion

### 3.1. Optical Performance of the TAG-Enabled System

One of the main novelties of our system is the capability to have fast and accurate control of the focus position of each color electronically, without the need for manual adjustments of the optics. It suffices to change the pulse delay between the pulses sent to each color of the RGB-LED and the TAG lens scanning, as shown in Figure 1b. Thus, the focal plane corresponding to each color can be independently selected within the axial range scanned by the TAG lens, –60 µm in current experiments. Note that the sinusoidal axial scanning of the TAG lens needs to be taken into consideration when selecting the delays.

Therefore, an initial calibration such as that used in Figure 1b is needed for each microscope configuration and TAG lens driving conditions, but once found, the delays remain stable over days. Importantly, the unique capability to select each color focal position independently allows for electronic control of chromatic aberrations. As shown in Figure 1c, the system point spread function (PSF) for each color can be adjusted to have an axial focus at user-selectable positions. Remarkably, this allows for compensating, enhancing, or even reversing the natural chromatic aberration of our microscope.

To quantitatively evaluate the optical performance of our system, we measured its spatial resolution using the Fourier Ring Correlation (FRC) analysis. FRC uses the correlation in Fourier space of two identically acquired images to extract a sample-dependent measurement of resolution [20]. In current experiments, we utilized a USAF resolution target (Photomask Portal, RTA39D22-434nm) as our sample and imaged it at a frame rate of 50 fps. We synchronized the TAG lens with each LED color to ensure that the native focal plane of the microscope objective was in focus for the green color. After color-splitting the captured images, we performed an FRC analysis of each color channel. Considering the 0.1 threshold criterion, the cut-off frequency values were 1.12, 1.37, and 1.41 µm^−1^ for the red, green, and blue channels, respectively (Figure 1d). They correspond to a spatial resolution of 893, 730, and 709 nm, which is close to the expected theoretical values based on the Rayleigh criterion—763 nm, 641 nm, and 573 nm for the red, green, and blue channels, respectively. The increase in the discrepancy between experimental and theoretical values can be attributed to the camera’s lower sensitivity at shorter wavelengths, with the blue channel being approximately 40% less sensitive than the red channel. Even though we attempted to partially compensate for this difference by adjusting the intensity of each LED color, we were unable to fully offset the effect. In any case, this is not an intrinsic limitation of the technique, and by using different camera models, diffraction-limited resolution could be achieved.

### 3.2. QPI with the TAG-Enabled System

As an initial test of our system’s ability to retrieve quantitative information from low-contrast samples, we reconstructed the phase of a cheek cell placed on a glass slide. In this case, the LED color pulses were synchronized with the TAG lens to focus the red at z = +5 µm, the green at z = 0 µm, and the blue at z = –5 µm. Following color splitting of a single-shot camera frame and correction for color leakage, we obtained the three axially displaced images required for TIE, as shown in Figure 2a. In experiments herein, we used a TIE solver based on the fast Fourier transform [2], but other TIE solvers lead to similar results. We used a spectrally-weighted mean wavelength, which has already been proven effective for partially coherent beams [10]. Notably, the reconstructed cheek cell’s phase map features a high contrast, in which distinct parts such as the cell membrane or nucleus are clearly distinguishable. This differs from the in-focus wide-field image (green channel), where structural details are difficult to discern. To benchmark our results, we reconstructed the phase map of the same sample using traditional TIE based on the sequential acquisition of three axially displaced monochrome images. Figure 2b shows the corresponding images captured by mechanically moving the sample at three different axial positions (+5.0 and −5.0 µm) with the TAG lens off and using only the green LED for pulsed illumination. The reconstructed phase map, computed using the same TIE solver as before, exhibits no significant differences compared to that obtained using our chromatic aberration-controlled system, only a 5% phase discrepancy at the central cell region. Importantly, the phase map obtained with our single-shot system provides at least a 3-fold enhancement in speed compared to sequential capturing, and this value can be significantly higher in practice when accounting for the time required for mechanical displacement and repositioning.

### 3.3. Proof of Principle: Tracking of a Paramecium

The intrinsic high speed of our TIE microscope, only limited by the frame rate of the camera, offers promise toward the characterization of dynamic or rapidly moving low-contrast samples. To validate this point, we tracked the movement of a water microorganism called Paramecium. These unicellular protozoans are easily cultivated and are commonly used as model organisms to study vesicle formation, cell nutrition, and digestive processes of the ciliate group [21]. Thanks to the cilia located at their outer membrane, Paramecia swim at high speed, which requires fast imaging systems to properly characterize their movement [22]. Additionally, fluorescent labelling is typically required to provide sufficient image contrast. As shown in Figure 3 and Appendix A, our TIE microscope provides a label-free alternative for characterizing the movement of a swimming Paramecium placed on a glass slide. In this case, the camera imaging rate was 150 fps, and the synchronization between color pulses and TAG lens was set to focus the red, green, and blue colors at focal planes located at z = +5 µm, z = 0 µm, and z = −5 µm, respectively. In each reconstructed phase map, the silhouette and nucleus of the paramecium can be clearly distinguished. Additionally, the retrieved phase values are in good agreement with the ones reported in the literature [23,24,25]. These results indicate that the system is sensitive to weak phase gradients of around 0.1 rad, corresponding to average refractive index variations of ~10^−2^ across micrometer-scale structures. Interestingly, it is possible to characterize the Paramecium movement as long as it remains within the field of view of the camera. Note that, compared to fluorescence imaging, the light dose required for an acceptable signal-to-noise ratio is low, which can help reduce phototoxicity. An evolution of the Paramecium’s position over time is shown in Figure 3b, where phase maps retrieved at different time instances are overlapped. A more refined analysis is shown in Figure 3c. In this case, we plotted the Paramecium’s nucleus position in each phase map using a single-particle tracking algorithm [26]. The trajectory of the microorganisms can be easily reconstructed with an impressive temporal resolution of 6.7 ms, only limited by the camera frame rate.

## 4. Conclusions

In conclusion, the use of an acousto-optic lens for chromatic aberration control enables quantitative phase imaging in a single camera snapshot. Leveraging the unique capabilities of the TAG lens and color-encoded pulsed illumination, our method achieves rapid and precise control of defocus distances, enabling the capture of the necessary images for TIE phase reconstruction in a single-color camera frame. As our results demonstrate, phase imaging at speeds of up to 150 phase reconstructions per second is possible, limited only by the camera frame rate. Such high speed, coupled with the dynamic selection of the defocus distances, is crucial for the label-free characterization of rapid events, such as the movement of a Paramecium, which would be difficult to capture using conventional phase imaging techniques. The possibility of implementing our approach in any wide-field microscope with minimal modifications can help expand the use of quantitative phase imaging techniques to applications where label-free and rapid imaging is key, such as the study of dynamic biological processes or moving microorganisms.

## Figures and Tables

**Figure 1 sensors-25-04503-f001:**
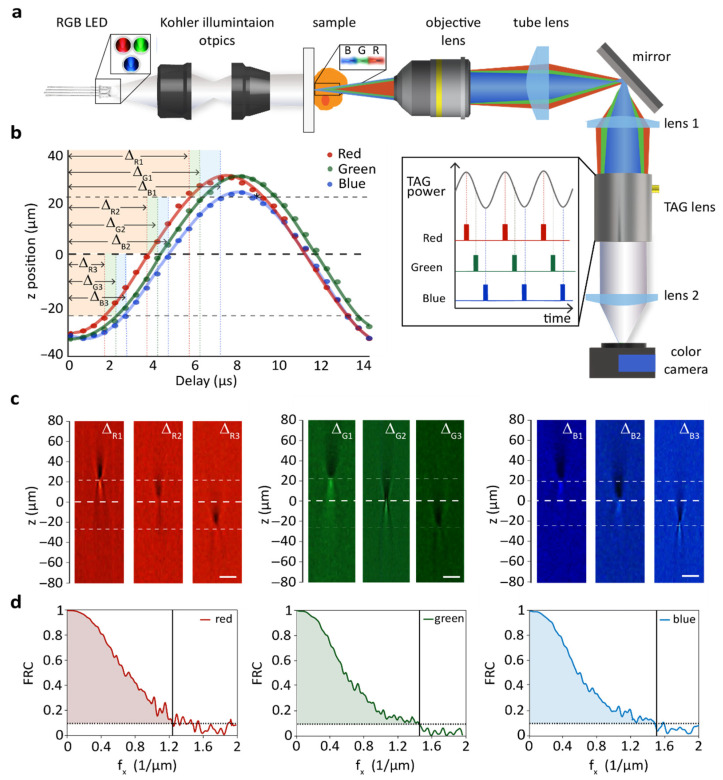
Principle of the TAG-enabled TIE microscope based on chromatic aberration control. (**a**) Schematic representation of the microscope and corresponding timing diagram. By controlling the TAG lens axial-scanning with pulsed multi-colored illumination, the focal plane corresponding to each color can be selected. (**b**) Plot of the microscope focal position versus different pulse delay values when illuminating with red, green, and blue light. (**c**) From left to right: Color-PSFs of the red, green, and blue channels at different time delays (see Figure 1b) measured using a 100 nm diameter microbead translated at different axial positions in steps of 2.5 µm. Scale bars 100 µm. (**d**) FRC measurements of the USAF target for red (left), green (middle), and blue (right) colors performed at 50 fps. The vertical solid black line indicates the cut-off frequency. The dashed lines indicate the 0.1 threshold criterion.

**Figure 2 sensors-25-04503-f002:**
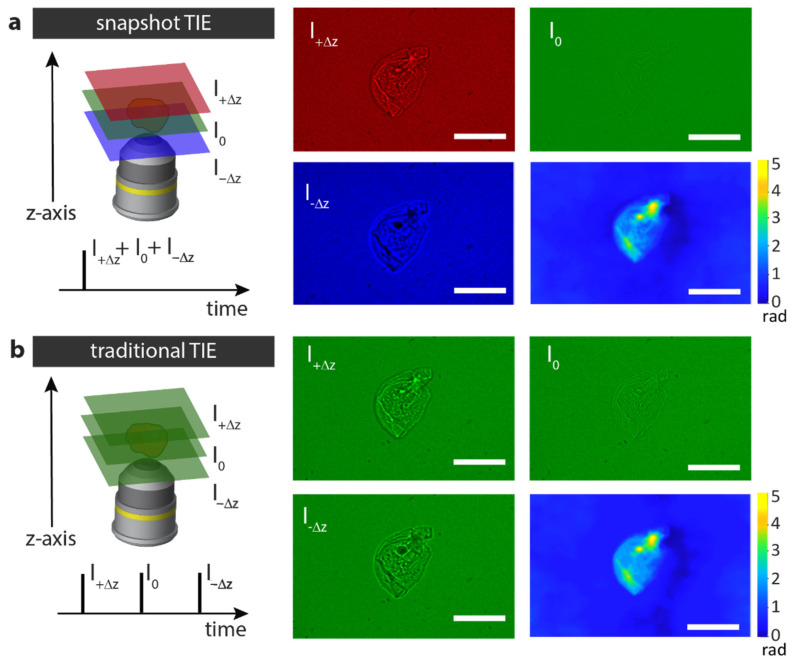
Quantitative phase imaging of a cheek cell. (**a**) Intensity images of a cheek cell captured with our chromatic aberration control system in a single snapshot, and the corresponding phase map retrieved by solving the TIE. The red, green, and blue channels, after color-splitting, correspond to axial planes z = +5 µm, z = 0 µm and z = −5 µm, respectively. (**b**) Intensity images of the same cheek cell and corresponding phase map captured sequentially using monochromatic illumination and mechanical translation of the sample at positions z = +5 µm, z = 0 µm and z = −5 µm. Scale bars 100 µm.

**Figure 3 sensors-25-04503-f003:**
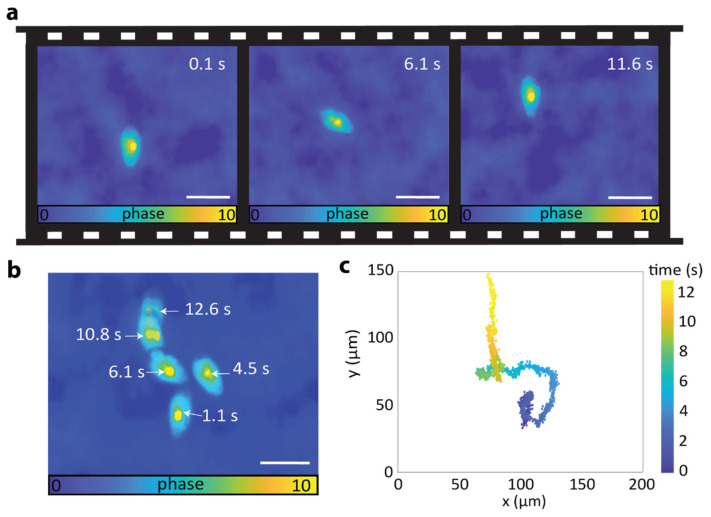
Rapid quantitative phase imaging of a moving Paramecium. (**a**) Phase maps of a Paramecium in water at different time instances were acquired at a rate of 150 phase maps/second. (**b**) Composite motion image of the Paramecium phase maps. (**c**) Plot of the Paramecium’s nucleus trajectory over time. Scale bars 100 µm.

**Table 1 sensors-25-04503-t001:** Measured intensity values (in percentage) for the different color channels after color-splitting an image captured when illuminating with a single-color LED before (left) and after color-leakage correction (right).

		Measured Camera Color (Raw)			Measured Camera Color (Correction)		
		Red (%)	Green (%)	Blue (%)	Red (%)	Green (%)	Blue (%)
Color LED	Red	68.5	28.3	3.3	99.7	0.3	0.0
	Green	5.3	69.3	25.4	1.3	97.8	0.9
	Blue	3.4	19.2	77.5	1.0	0.7	98.3

## Data Availability

The data that support the findings of this study are available from the corresponding author upon reasonable request.

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
