# Peer review of "Snapshot Quantitative Phase Imaging with Acousto-Optic Chromatic Aberration Control"

_sensors, 2025, doi:10.3390/s25144503_

Round 1

Reviewer 1 Report

Comments and Suggestions for Authors

The manuscript entitled, "Snapshot quantitative phase imaging with acousto-optic chromatic aberration control," provides a novel fast method for performing quantitative phase imaging, which is useful for inspecting unstained microscopic samples. I find the manuscript interesting and recommend acceptance after the following revisions.

(1) Could an iterative phase retrieval algorithm like Gerchberg Saxton be used to recover the phase instead of using the transport of intensity equation? What are the advantages and disadvantages of each reconstruction approach when using the TAG-enabled images?

(2) The authors mention that they use a USAF target to characterize the resolution of their system. The authors should show these images of the USAF target in the manuscript so that readers can easily see the resolution. Also, if the authors are using a USAF target, why not just quote the smallest set of lines that are visually resolvable rather than using the more complex Fourier Ring Correlation analysis?

Author Response

We would like to thank all the reviewer for the accurate work of revision and for the useful comments and suggestions, which helped us to improve the quality of the manuscript. Please, find below a list of point-by-point answers (in black) to his/her comments (in blue) along with the changes performed in the revised version of the manuscript (in red).

(1) Could an iterative phase retrieval algorithm like Gerchberg Saxton be used to recover the phase instead of using the transport of intensity equation? What are the advantages and disadvantages of each reconstruction approach when using the TAG-enabled images?

We thank the reviewer for this valuable question. Iterative phase retrieval algorithms such as the Gerchberg–Saxton (GS) method can, in principle, be used to recover phase information from TAG-enabled axial images. Traditionally, GS requires coherent illumination and two intensity measurements at different planes. However, recent work by Anand et al. [new ref. 3] has shown that the GS algorithm can be adapted for use with partially coherent or even incoherent light sources, including LEDs, and implemented in a lensless QPI configuration.

Despite this flexibility, such methods face key limitations in real-time biological imaging scenarios. First, even with convergence in only a few iterations, the GS algorithm is inherently iterative and computationally more intensive than the Transport of Intensity Equation (TIE), which is non-iterative and directly reconstructs the phase under the weak phase approximation. Second, the GS method often requires precise calibration of the propagation distances, careful phase unwrapping, and may be sensitive to noise, particularly when used with incoherent sources. Additionally, real-time multi-plane imaging using GS is difficult, as it typically requires multiple exposures or propagation models tailored to a fixed imaging geometry.

In contrast, the TIE is well-suited to our TAG-enabled chromatic-aberration-controlled microscope, which naturally acquires high-speed z-stacks using incoherent illumination in a single pass. TIE leverages these axial intensity gradients to provide quantitative phase maps with minimal computational load and is more robust for dynamic, live imaging conditions.

We have included the following text to the main manuscript to highlight the advantages of TIE with respect to the GS algorithm for our system and added new reference 3.

While iterative phase retrieval algorithms such as the Gerchberg-Saxton (GS) method can be applied in similar configurations, they are computationally intensive, sensitive to noise, and less suited for real-time imaging [3]. In contrast, TIE enables non-iterative phase reconstruction from axial intensity gradients, making it more computationally efficient, robust to noise, and overall, more appropriate for dynamic biological imaging.

[3] Anand V, Katkus T, Linklater DP, Ivanova EP, Juodkazis S. Lensless Three-Dimensional Quantitative Phase Imaging Using Phase Retrieval Algorithm. J Imaging. 2020 Sep 20;6(9):99. doi: 10.3390/jimaging6090099. PMID: 34460756; PMCID: PMC8321078.

(2) The authors mention that they use a USAF target to characterize the resolution of their system. The authors should show these images of the USAF target in the manuscript so that readers can easily see the resolution. Also, if the authors are using a USAF target, why not just quote the smallest set of lines that are visually resolvable rather than using the more complex Fourier Ring Correlation analysis?

We believe reporting microscope resolution using FRC is more rigorous and objective than simply stating the smallest resolvable feature on a USAF target. FRC quantifies the true, data-dependent resolution on an image based on frequency content, and it reflects how well signal is preserved across spatial frequencies in a real image, not just under ideal test conditions. On the contrary, the resolution given by the smallest line pair pattern of a USAF target that can be distinguished by eye is more subjective and it is influenced by contrast and illumination. As a consequence, resolution cannot be properly quantified. For these reasons, we believe that the FRC analysis shown in the article is more than sufficient to properly characterize the resolution of our system.

Reviewer 2 Report

Comments and Suggestions for Authors

 12 June 2025

Reviewer’s comments on „Snapshot quantitative phase imaging with acousto-optic chromatic aberration control (Sensors)” by C.  Alexandropoulos et al.

The work is about the establishing a new imaging approach utilizing phase determination by solving the transport of intensity equation (TIE). The curiosity of this approach is that only two images (a focused one and a defocused one) are needed instead of three ones (a focused one and two defocused ones). This is realised by pulsed excitation with three light beams of different colours (red, green, blue) and control of detecting focus position by a fluid lens, a TAG lens for acousto-optic chromatic aberration control. The point is achieving high speed phase mapping for imaging dynamic processes like motion of cells.

The aim is quite interesting and up-to-date and the whole work has been realised well. The way of presentation is also good. I have only two subjectal concerns and have found four formal shortcomings.

Subjectal concerns:

  1. The main concern arisen in me is the behaviour of error propagation law during solving TIE. What should be the maximum sinal/noise ratio (quantified as the ratio of the size of useful signal and the size of background) for achieving adequatly sharp images. To put it in another way, from the viewpoint of a phase sensor, because the aim of phase detection is to render visible which is otherwise is invisible to the human eye: what is the „maximum degree of transparency” wich is detectable by this method? „Maximum transparency” is quantified here as a minimum change in refractive index or width of a phase object.

  1. Autofluorescence in the forms of phosphorescence and delayed fluorescence might be disturbing because of the extremely long decay times comparable to the time delays used here.      

Formal concerns

  1. Please define QPI and TAG in words at their first occurrence.
  2. Is the „pulse delay generator (DG645)” commonly called also „lock-in amplifier”? If it is, it might be mentioned in the text, becuase of the wide-spread use of this term.
  3. Please specify the bandwidth of the three LEDs in nm, for the sake of appreciating the colour-leakage error.
  4. In the references 24 and 25 please correct the „author name repitition errors”: Zhang M, Ma Y, Zheng J, Liu L, Gao P occuring more than once in reference 24 and similarly Zhao X, Wang Y, Li D in reference 25.

Author Response

We would like to thank the reviewer for the accurate work of revision and for the useful comments and suggestions, which helped us to improve the quality of the manuscript. Please, find below a list of point-by-point answers (in black) to his/her comments (in blue) along with the changes performed in the revised version of the manuscript (in red).

Subjectal concerns:

1. The main concern arisen in me is the behaviour of error propagation law during solving TIE. What should be the maximum signal/noise ratio (quantified as the ratio of the size of useful signal and the size of background) for achieving adequatly sharp images. To put it in another way, from the viewpoint of a phase sensor, because the aim of phase detection is to render visible which is otherwise is invisible to the human eye: what is the „maximum degree of transparency” wich is detectable by this method? „Maximum transparency” is quantified here as a minimum change in refractive index or width of a phase object.

We thank the reviewer for this valuable question. While a full theoretical error propagation analysis is beyond the scope of this manuscript, from our experimental results (Figs. 2 & 3) we demonstrate that our system reliably reconstructs phase shifts from low-contrast samples such as cheek cells where features like membranes and nuclei were clearly resolved. These results indicate that the system is sensitive to weak phase gradients of 0.1 rad., which corresponding to refractive index variations of ~10-2 across micrometer-scale structures. For example, in the case of a freely swimming paramecium in water, from the literature we know that the refractive index varies from 1.31 to 1.37 ( References 25 and 26). Additionally, the close match to conventional TIE reconstructions (with <5% difference in the cell area) supports the stability of the phase retrieval process despite the inherently lower SNR of color-split images.

We have added the following information in the main text:

These results indicate that the system is sensitive to weak phase gradients of around 0.1 rad, corresponding to average refractive index variations of ~10-2 across micrometer-scale structures.

2. Autofluorescence in the forms of phosphorescence and delayed fluorescence might be disturbing because of the extremely long decay times comparable to the time delays used here.      

While autofluorescence cannot be discarded, we would like to point out that we are collecting the transmitted light that passes through our sample without any dichroic mirror – only the Bayer filters of the camera.  Given that autofluorescence is a weak process, we believe it does not play any significant role in our experiments. In fact, we did not observe any imaging artifacts consistent with this phenomenon.

Formal concerns

1. Please define QPI and TAG in words at their first occurrence.  We thank the reviewer for this insightful comment. They have been defined in lines 50 and 57-58.

2. Is the „pulse delay generator (DG645)” commonly called also „lock-in amplifier”? If it is, it might be mentioned in the text, becuase of the wide-spread use of this term. The pulse delay generator (DG645) is not a lock-in amplifier. As its name indicates, it is an electronic device that generates precise delays. In our setup, it is used to generate precise timing signals and control the time delays for each color channel in synchronization with the TAG lens.

3. Please specify the bandwidth of the three LEDs in nm, for the sake of appreciating the colour-leakage error. We thank the reviewer for this valuable comment. However, we believe that the issue of color leakage has been effectively addressed through our experimental design and data processing, as evidenced by the results presented in Table 1. These results demonstrate minimal cross-talk and reliable separation between the channels, supporting the robustness of our approach.

 4. In the references 24 and 25 please correct the „author name repitition errors”: Zhang M, Ma Y, Zheng J, Liu L, Gao P occuring more than once in reference 24 and similarly Zhao X, Wang Y, Li D in reference 25.  We thank the reviewer for this insightful comment. The text has been modified with the corrected references.